# Syn3DTxt: Embedding 3D Cues for Scene Text Generation

## Abstract

*This study aims to investigate the challenge of insufficient three-dimensional context in synthetic datasets for scene text rendering. Although recent advances in diffusion models and related techniques have improved certain aspects of scene text generation, most existing approaches continue to rely on 2D data, sourcing authentic training examples from movie posters and book covers, which limits their ability to capture the complex interactions among spatial layout and visual effects in real-world scenes. In particular, traditional 2D datasets do not provide the necessary geometric cues for accurately embedding text into diverse backgrounds. To address this limitation, we propose a novel standard for constructing synthetic datasets that incorporates surface normals to enrich three-dimensional scene characteristic. By adding surface normals to conventional 2D data, our approach aims to enhance the representation of spatial relationships and provide a more robust foundation for future scene text rendering methods. Extensive experiments demonstrate that datasets built under this new standard offer improved geometric context, facilitating further advancements in text rendering under complex 3D-spatial conditions.*

## 1. Introduction

Recent advances in scene text generation have enabled remarkable progress in synthesizing text-rich images through image-to-image and text-to-image paradigms [1, 2, 7, 9, 13, 16]. However, a critical bottleneck persists: existing methods predominantly rely on training data confined to 2D planar text (e.g., book covers, posters)(see Fig. 1a) or synthetic benchmarks inherited from SRNet-style pipelines [11, 14, 16](see Fig. 1b). While these datasets suffice for frontal-view text rendering, they fundamentally lack the intricate 3D visual effects ubiquitous in real-world scenarios—such as perspective distortion, multi-axis rotations, and complex scene text arrangement.This discrepancy significantly restricts the model's generalizability in practical applica-

tions. Consequently, it exhibits reduced accuracy in text recognition and editing across diverse real-world environments, along with suboptimal image quality in scene text generation.

Current approaches face two intertwined limitations. First, while real-world datasets [1, 3] (see Fig. 1a) encompass 3D text scene data, they suffer from sparse text instances, inconsistent annotation quality, and insufficient diversity, leading to significant shortcomings in robust training. Moreover, these datasets are primarily designed for scene text recognition tasks, providing only bounding box annotations without 3D characteristics labeling, which hinders the model's ability to learn complex spatial relationships and realistic text placements. Second, existing synthetic datasets [14] predominantly employ simplified 2D warping strategies, failing to effectively simulate the geometric interactions between text and 3D scenes in a physically plausible manner. Although some studies [4, 8] attempt to generate text that aligns with the 3D layout and color of the background, these data sets are still mainly constructed for text recognition and lack complete 3D annotations. Consequently, even state-of-the-art models [11, 16] continue to struggle with tasks requiring perspective consistency, text placement in non-frontal viewpoints, or maintaining realistic background textures on curved surfaces.

To fully address these challenges, we propose a novel synthetic data generation engine that directly embeds 3D geometric characteristics into text masks, improving the model's understanding of text-scene interactions. Compared to previous approaches that encode only simplistic 2D positional maps [14], our primary innovation lies in the representation of 3D spatial characteristics, such as surface normals, by RGB-colored masks, providing the model with more intuitive geometric cues. This enables accurate learning of text-environment interactions under precise perspective projections. Specifically, we render highly detailed 3D text meshes with fine-grained control over background, text content, curvature, color, 3D orientation, and font design, ensuring both diversity and realism in the generated data. This text data generation engine offers two key advantages: (1) it disentangles

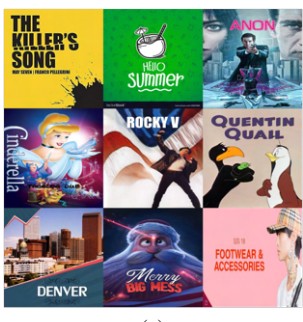
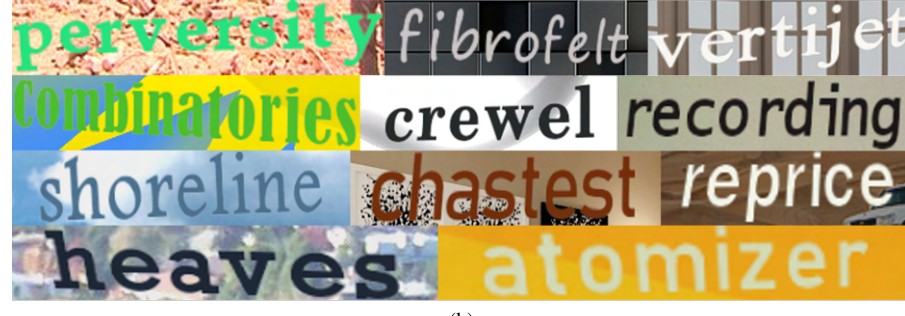

(a)         (b)

Figure 1. Example of previous Dataset (a) MARIO-10M, constructed by [1], which captures real-world text instances predominantly within 2D imagery but lacks comprehensive 3D geometric annotations. (b) Synthetic dataset generated using the SRNet[14] pipeline, which primarily applies simplified 2D warping transformations without incorporating 3D spatial details. These examples illustrate that existing datasets mainly consist of 2D images and rarely include accurate representations of text within realistic 3D environments, limiting their utility in training robust models capable of handling complex spatial interactions in scene text synthesis tasks.

complex geometric transformations (such as perspective foreshortening, scaling, and rotation) from appearance features, allowing for more precise geometric reasoning; and (2) it provides physically grounded supervision cues, ensuring that text is realistically embedded into diverse 3D scenes while adhering to real-world lighting and geometric constraints.

We validate the efficacy of our method rigorously using extensive benchmarking experiments on the MOSTEL architecture. Experimental results demonstrate that models trained on our proposed 3D-augmented dataset outperform traditional 2D baselines by achieving an impressive 15% improvement in perspective-consistent text editing, as quantified by Perspective-Aware *SSIM*, and 17.7% in *FID* [5]. Qualitative assessments (Figure 2) further substantiate our approach's superiority, exhibiting enhanced realism and precision, especially in challenging scenarios involving oblique angles, curved surfaces, and complex lighting conditions. To encourage widespread adoption and facilitate future research endeavors, we will publicly release our data generation toolkit along with pre-trained models.

Our contributions are summarized as follows:

- Introduce a synthetic data generation framework with 3D geometric cues and controllable variations, and publicly release the toolkit to support future research.
- Release two novel synthetic datasets, Syn3DTxt and Syn3DTxt-wrap, specifically designed for scene text rendering. These datasets explicitly incorporate 3D geometric supervision to facilitate the training of perspective-aware text editing models.
- Experimental validation demonstrates a 15% improvement in SSIM and 17.7% in FID for perspective-consistent text editing tasks compared to traditional 2D methods.

This work can provide a novel perspective to the research on scene text generation. The code and dataset are available at: https://github.com/xxxxxxx-123456789/Syn3DTxt

In the following, we first review previous work in Sec. 2, then present our approach in Sec. 3, then the experiments in Sec. 4, and then a conclusion to this work in Sec. 5.

## 2. Related Work

The field of scene text editing has long been explored, with many studies and synthetic dataset generation methods proposed. However, the challenge lies in accommodating the angular variations present in three-dimensional environments. Building on this foundation, our work provides a generator capable of producing synthetic data with text orientation vectors, which can be used for training text replacement models. In the following, we discuss the relationship between our work and several related research areas.

### 2.1. Real Datasets

Real datasets continue to play an essential role in benchmarking and validating scene text models. Datasets such as CUTE80[12] provide curved text instances that challenge recognition systems with their non-linear structures. Total-Text offers a comprehensive set of arbitrarily oriented text instances, which are particularly useful for evaluating detection models under diverse conditions. Additionally, MARIO-10M[1] serves as a large-scale real dataset that further aids in assessing the generalization and robustness of models in real-world scenarios. These real datasets complement synthetic data by introducing the natural variations and complexities that

occur in practical applications, ensuring that the developed models are capable of handling diverse text appearances and environmental conditions.

## 2.2. Synthetic Data

In recent years, due to the high cost and potential errors associated with manually annotating scene text data, synthetic data has played a crucial role in text detection and recognition. For example, the Synth90k[6] dataset contains 9 million synthetic text instance images generated from 90k common English words. These words are rendered onto natural images using random transformations and effects, such as various fonts, colors, blurs, and noise, and every image is annotated with a ground-truth word. This dataset effectively emulates the distribution of text images from real scenes and serves as an excellent substitute for real-world data when training data-hungry deep learning algorithms.

Moreover, in the field of scene text recognition, SynthTIGER[15] presents a synthesis engine that integrates effective rendering techniques from existing methods (such as Synth90k[6] and SynthText[4]) to produce bounding boxes for text images that incorporate both text noise and natural background noise. SynthTIGER[15] overcomes the long-tail distribution problem inherent in traditional synthetic datasets by introducing two strategies: text length distribution augmentation and infrequent character augmentation. These techniques balance the distribution across different text lengths and character frequencies, thereby enhancing the generalization ability of scene text recognition models.

Additionally, SynthText3D[8] leverages characteristic from 3D virtual worlds to synthesize scene text images, diverging from traditional methods that simply paste text onto static 2D backgrounds. Based on Unreal Engine 4 and the UnrealCV plugin, SynthText3D employs four modules—Camera Anchor Generation, Text Region Generation, Text Generation, and 3D Rendering to integrate realistic perspective transformations, illumination variations, and occlusion effects. As a result, the generated images more accurately reflect the complexity of real-world environments. Together, these studies demonstrate the significant potential of synthetic data to emulate real-world scene text distributions and diverse visual effects.

## 2.3. Scene Text Editing

Beyond synthetic data generation, scene text editing, where text replacement, content modification, and style preservation are critical challenges, has also attracted increasing attention recently. SRNet (Editing Text in the Wild)[14], proposed by Liang Wu et al., is the first end-to-end trainable network addressing scene text editing at the word level. Its architecture decomposes the text editing task into three main components: the text conversion module, the background inpainting module, and the fusion module. The text conversion module transfers the text style from a source image to the target text while preserving the text skeleton through skeleton-guided learning to maintain semantic consistency. The background inpainting module restores the background in the text regions. The fusion module then integrates these outputs to generate visually realistic and stylistically consistent edited images. Notably, SRNet[14] also introduces a synthetic data generator that randomly selects fonts, colors, and deformation parameters to render text on background images in a unified style while automatically producing corresponding background, foreground text, and text skeleton annotations via image skeletonization, thereby providing large scale synthetic training data.

In addition, MOSTEL (Exploring Stroke-Level Modifications for Scene Text Editing)[11] further investigates stroke-level modification techniques by generating explicit stroke guidance maps. This approach effectively differentiates and preserves unchanged background regions while focusing on editing rules within text areas. MOSTEL[11] combines this with semi-supervised hybrid learning, leveraging extensive synthetic annotated data alongside unlabeled real-world images to bridge the domain gap between synthetic and real data. Experimental results indicate that MOSTEL[11] outperforms previous methods in various quantitative metrics.

Furthermore, TextCtrl (Diffusion-based Scene Text Editing with Prior Guidance Control)[16] is a diffusion-based method centered on content modification and style preservation. It addresses common issues found in GAN-based and diffusion-based STE methods by constructing fine-grained text style disentanglement and robust text glyph structure representations. TextCtrl[16] explicitly incorporates style-structure guidance into its model design and training, significantly improving text style consistency and rendering accuracy. Additionally, it introduces a Glyph-adaptive Mutual Self-attention mechanism to further leverage style priors, enhancing style consistency and visual quality during inference. To fill the gap in real-world STE evaluation, the authors also created the first real-world image-pair dataset, Scene-Pair, which facilitates fair comparisons. Experimental results demonstrate that TextCtrl[16] outperforms prior methods in both style fidelity and text accuracy.

## 3. Methodology

Most text synthesis studies focus on generating text within 2D imagery [6, 14, 15] but struggle to capture the complex geometric interactions between text and real-world 3D environments (refer to Fig. 1). Although some

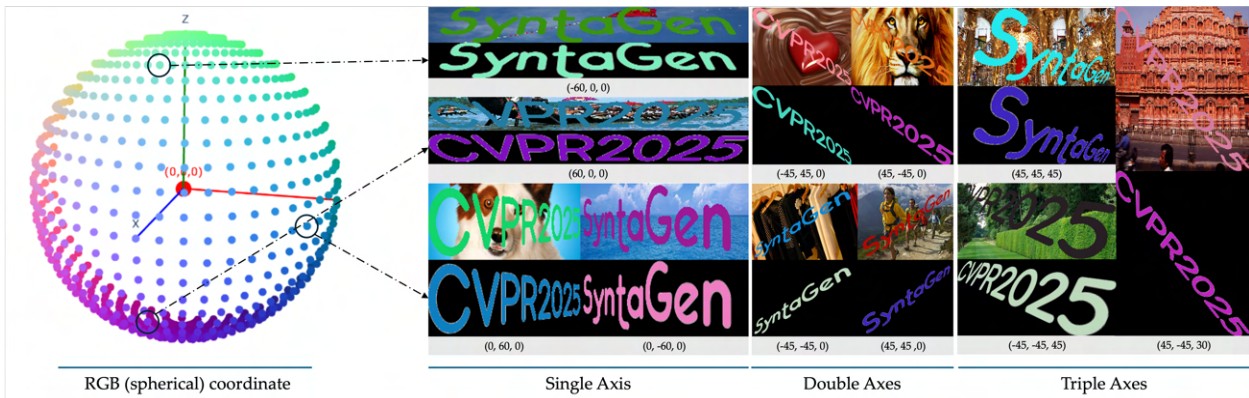

Figure 2. Visualization of RGB-encoded normal vectors within a spherical coordinate system. Each point on the sphere represents a distinct orientation, with its normal vector coordinates mapped directly to RGB colors. By connecting these spherical points to corresponding text images generated at specific rotation angles, we illustrate how text rendering outcomes vary according to precise 3D orientations. All angles follow the defined order $(\theta, \phi, \gamma)$.

| Number of Axes | Single | | | Dual | | | Triple |
|---|---|---|---|---|---|---|---|
| | $(\phi)$ | $(\theta)$ | $(\gamma)$ | $(\theta,\phi)$ | $(\theta,\gamma)$ | $(\phi,\gamma)$ | $(\theta,\phi,\gamma)$ |
| Percentage (%) | 20% | 20% | 20% | 20% | 5% | 5% | 10% |

Table 1. Distributions of rotation angles in terms of single-, dual-, and triple-axis combinations, reflecting realistic rotational behavior observed in real-world scenarios.

| Rotate Angle | Catagory | | |
|---|---|---|---|
| | Small | medium | large |
| CCW (°) | 30° | 45° ∼ 60° | 65° ∼ 70° |
| CW (°) | −30° | −45° ∼ −60° | −65° ∼ −70° |

Table 2. Categorization of rotation angles into small, medium, and large, further subdivided into (CW) and (CCW) rotations.

work attempts to integrate text into 3D scenes [4, 8], they primarily serve as data augmentation for text recognition and lack comprehensive 3D geometric details to guide generative models in learning perspective variations. Instead of designing new model architectures to tackle real-world challenges, we focus on 3D feature augmentation based on object attributes, providing novel insights to improve model interpretability and scene text generation quality. The following sections present our object attribute editing tool and the Syn3DTxt dataset, highlighting their significance in scene text synthesis.

## 3.1. Controlling text, 3D orientation and curvature

In general, human visual system exhibits remarkable robustness to changes in position, orientation, and viewpoint. However, it remains an open question whether deep learning models can consistently handle variations in these object properties. To investigate this issue, we propose a data generation pipeline that manipulates images by controlling the 3D orientation and curvature of objects, thereby evaluating model performance under realistic visual transformations.

The process is as follows. First, a fixed-size text mask image is generated based on the provided textual content and font, with its initial state represented as a two-dimensional plane $P \in \mathbb{R}^{3 \times h \times w}$ next, a uniform two-dimensional arc distortion is applied to induce varying degrees of curvature in the text image. Subsequently, to more faithfully simulate spatial variations encountered in real-world scenes, a 3D rotation transformation is imposed on the text image. This transformation encompasses single-axis, dual-axis, and triple-axis rotations along the X, Y, and Z axes (corresponding to roll $\gamma$, pitch $\theta$, and yaw $\phi$, respectively), thus mimicking the diversity and complexity of objects in practical scenarios and generating $T_x \cdot P$, $T_y \cdot P$, and $T_z \cdot P$. (see Eqs. (1) to (3), in which $T_x$, $T_y$, $T_z$ denote the rotation matrices corresponding to rotations about the X, Y, and Z axes, respectively. Specifically, $T_x$ adjusts the roll ($\gamma$), $T_y$ modifies the pitch ($\theta$), and $T_z$ alters the yaw ($\phi$) of the text mask $P$. When these matrices are applied to $P$, they generate rotated versions of the text, simulating a range of real-world 3D perspective variations.)

$$T_x = \begin{bmatrix} \cos\gamma & -\sin\gamma & 0 & 0 \\ \sin\gamma & \cos\gamma & 0 & 0 \\ 0 & 0 & 1 & 0 \\ 0 & 0 & 0 & 1 \end{bmatrix} \quad (1)$$

$$T_y = \begin{bmatrix} \cos\phi & 0 & -\sin\phi & 0 \\ 0 & 1 & 0 & 0 \\ \sin\phi & 0 & \cos\phi & 0 \\ 0 & 0 & 0 & 1 \end{bmatrix} \quad (2)$$

$$T_z = \begin{bmatrix} 1 & 0 & 0 & 0 \\ 0 & \cos\theta & -\sin\theta & 0 \\ 0 & \sin\theta & \cos\theta & 0 \\ 0 & 0 & 0 & 1 \end{bmatrix} \quad (3)$$

Since matrix operations are not commutative (i.e., $AB \neq BA$), the order of rotations must be rigorously defined during multiaxis transformations to accurately replicate real-world viewpoint changes. In practice, humans typically maintain a view cone, and scene texts, such as signboards, are often placed with a fixed roll $\gamma$. We thus design that the rotation in roll $\gamma$ should take place before the rotations taken place in pitch $\theta$ and yaw $\phi$. Moreover, when simulating viewpoint changes solely through rotations (as opposed to translations), it is critical to determine whether to adjust the vertical rotation pitch $\theta$ or the horizontal rotation yaw $\phi$ first. For instance, close-up viewpoint where vertical displacement is more pronounced, adjusting pitch $\theta$ first enables rapid alignment with the object, followed by fine-tuning with yaw $\phi$; in contrast, for distant signboards, where mathematically tends toward zero as distance increases and vertical angular effects become minimal, the influence is predominantly governed by horizontal parallax, thus necessitating the prioritization of yaw $\phi$ (refer to Eq. (4), in which y represents the height and x represents the distance.)

$$\lim_{x \to \infty} \tan^{-1}\left(\frac{y}{x}\right) \approx 0 \quad (4)$$

Additionally, in contrast to simply rotating the entire plane, we have also generated text data with three-dimensional bending, in which each character exhibits a distinct normal vector (see Fig. 3). This approach more faithfully captures the complex and varied transformations of objects as encountered in real-world scenes.

In summary, by carefully defining the sequence of multiaxis rotations based on the target object's relative position and displacement within the field of view, our approach closely emulates the variations in real-world observation. This enables a more precise evaluation of the robustness of deep learning models when faced with such visual changes.

## 3.2. Syn3DTxt Dataset

With the aforementioned methods, we generate text images based on a large-scale text corpus and a diverse font library, incorporating arc distortion, font transformation, and 3D rotation processing. Precise mask annotations are provided for each pair of generated images. To ensure the quality of the dataset, we selected 70 fonts from

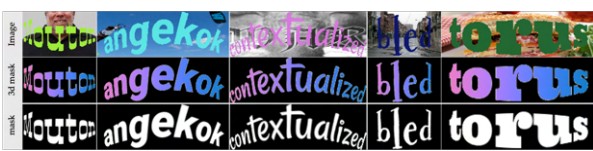

Figure 3. Example of generated text data with three-dimensional bending effects. The first column shows the rendered text images; the second column displays the corresponding normal vector masks encoded in RGB, highlighting detailed 3D spatial characteristics; and the third column presents binary masks indicating text regions. Unlike simple planar rotations, our approach assigns distinct normal vectors to each character, enabling more accurate modeling of the complex geometric transformations commonly observed in real-world scenes.

a curated font collection to guarantee that the rendered text is both clear and aesthetically pleasing. Ultimately, our dataset comprises over 200k paired training samples and 6k testing samples generated from the initial text files, with each sample undergoing both arc distortion and 3D rotation to fully simulate the diverse variations of text in natural scenes.

For 3D rotation processing, we defined a rotation distribution to realistically mimic object rotations observed in real-world scenarios. Specifically, the designed rotation distribution includes (see Tab. 1):

**Single-axis rotations:** rotations around the $\theta$, $\phi$, and $\gamma$ axes each account for 20%, ensuring balanced representation of each axis;
**Dual-axis rotations:** the $\theta + \phi$ combination comprises 20%, while the $\theta + \gamma$ and $\phi + \gamma$ combinations each comprise 5%. This reflects real-world scenarios where horizontal and vertical rotations ($\theta$ and $\phi$) dominate, while other combinations occur less frequently;
**Triple-axis rotations:** rotations involving all three axes ($\theta + \phi + \gamma$) constitute 10%, adding further complexity to the data set.

Additionally, based on visual inspection after coordinate calculations, we categorized the rotation angles into small, medium, and large, further subdividing them into clockwise and counterclockwise rotations (see Tab. 2; CCW denotes counterclockwise rotation, CW denotes clockwise rotation). To intuitively visualize normal vectors, we mapped the calculated coordinates to RGB color space (see Fig. 2 and Eq. (5)). This approach enhances the rotational diversity of the data set, providing comprehensive and varied training data to ensure robust model performance.

$$\begin{bmatrix} R \\ G \\ B \end{bmatrix} = \begin{bmatrix} \sin\theta \times \cos\phi \\ \sin\theta \times \sin\phi \\ \cos\theta \end{bmatrix} \quad (5)$$

To further simulate the appearance of curved text in natural scenes, each pair of text images is also randomly subjected to three arc distortion operations (namely 0°, 60° and 120°). This dual transformation strategy not only preserves the integrity of the original text characteristic but also introduces a controlled degree of deformation, making the generated dataset more suitable for training text generation models that can handle diverse scene requirements.

## 4. Experiments

To validate the effectiveness of our proposed method, we conducted extensive experiments utilizing our novel synthetic datasets integrated with detailed surface normal. We adopted the MOSTEL architecture [11] as a baseline, modifying its decoder output from a single channel (1D) to three channels (3D). This modification enables the model to directly leverage the richer geometric characteristic encoded in the RGB masks. We evaluated the impact of our proposed 3D-augmented datasets on scene text editing tasks through comprehensive experimentation.

### 4.1. Datasets

Considering the lack of publicly available benchmarks explicitly tailored for 3D-enhanced scene text editing, we introduced the **Syn3DTxt** dataset, specifically designed to address this gap.

**Syn3DTxt.** Our proposed synthetic data set comprises 150,000 images, meticulously generated using our advanced methodology. Each image integrates explicit 3D surface normal via RGB masks that encode precise surface normals. We utilized 70 high-quality fonts and various transformations, including random rotations, curvature alterations, and multiaxis spatial transformations, to realistically emulate complex real-world scenarios. Furthermore, two specialized data sets for evaluation, **Syn3DTxt-eval-2k** and **Syn3DTxt-eval-advanced**, each containing 2,000 images, are included for complete evaluation. Notably, **Syn3DTxt-eval-advanced** specifically contains images featuring medium- and large-angle rotations, categorized according to the criteria detailed in Tab. 1.

**Syn3DTxt-wrap-2k.** To further evaluate performance in scenarios involving pronounced three-dimensional bending (see Fig. 3), we generated an additional 2,000 images with increased complexity and varied curvature transformations. This subset facilitates assessing the model's capacity to handle intricate geometric distortions. This test set will be used to further evaluate our method.

**MOSTEL-150K.** The dataset comprises 150,000 labeled synthetic images, specifically generated for supervised training of the MOSTEL method. Each image is created by integrating various randomized visual transformations applied across 300 distinct fonts and 12,000 diverse background images.

**Tamper-Syn2k.** The Tamper-Syn2k dataset, introduced by [11] in their work on stroke-level modifications for scene text editing, addresses the scarcity of public evaluation data sets in the field of Scene Text Editing (STE). It comprises 2,000 pairs of synthetic images, each pair maintaining consistent style attributes such as font, size, color, spatial transformation, and background. However, Tamper-Syn2k exhibits limited diversity in perspective and curvature transformations, which may restrict models' ability to generalize to real-world scenarios involving complex viewing angles and text curvatures.

**MLT-2017.** The ICDAR 2017 Multilingual Scene Text [10] dataset comprises diverse images of real-world scene text covering multiple scripts and languages, including Arabic, Chinese, English, and others. It consists of 34,625 images annotated with text transcripts, offering a valuable resource for training robust scene-text methods. Specifically, it was used for the training of MOSTEL, enhancing its effectiveness in practical multilingual scenarios.

### 4.2. Training Strategy

To accommodate the richer geometric representations provided by our 3D masks, the MOSTEL decoder was modified to output predictions with three channels instead of the original single channel. This modification served as the basis for our structured, incremental training strategy, designed to progressively introduce and reinforce complex 3D geometric characteristic within the MOSTEL architecture.

We structured our training strategy into three distinct phases:

1. **Baseline Training.**: We initialized the model with the original 150,000-image MOSTEL synthetic dataset (MOSTEL-150k) and the 34,625-image real-world scene text dataset (MLT-2017). Both datasets are characterized by planar 2D masks, establishing a foundational baseline for the model's capabilities.
2. **3D Feature Augmentation.**: Subsequently, the model was fine-tuned using our proposed Syn3DTxt-150k dataset, integrating detailed surface normal via surface normal RGB masks. This step further enhanced the model's spatial awareness and depth perception.

| Models | Syn3DTxt-eval-2k | | | | Syn3DTxt-wrap | | | | Syn3DTxt-eval-advanced | | | | Tamper-Syn2k | | | |
|---|---|---|---|---|---|---|---|---|---|---|---|---|---|---|---|---|
| | PSNR ↑ | SSIM ↑ | MSE ↓ | FID ↓ | PSNR ↑ | SSIM ↑ | MSE ↓ | FID ↓ | PSNR ↑ | SSIM ↑ | MSE ↓ | FID ↓ | PSNR ↑ | SSIM ↑ | MSE ↓ | FID ↓ |
| SRNet [14] | 17.011 | 0.5283 | 0.0234 | 80.502 | 16.433 | 0.5027 | 0.0267 | 61.832 | 17.152 | 0.5259 | 0.0228 | 87.333 | 18.042 | 0.6114 | 0.0216 | 51.538 |
| TextCtrl [16] | 17.837 | 0.6067 | 0.0293 | 36.288 | 16.646 | 0.5371 | 0.0266 | 40.990 | 18.523 | 0.6302 | 0.0188 | 34.800 | | | | |
| MOSTEL† [11] | 20.527 | 0.7265 | 0.0119 | 40.005 | 17.386 | 0.6185 | 0.0179 | 45.630 | 19.855 | 0.7677 | 0.0133 | 41.311 | **20.489** | **0.7912** | **0.0128** | **36.337** |
| MOSTEL + 2D Finetuned | 20.846 | 0.7215 | 0.0103 | 33.991 | 17.196 | 0.6000 | 0.0188 | 37.625 | 21.356 | 0.7651 | 0.0114 | 34.803 | 19.746 | 0.6666 | 0.0157 | 41.921 |
| MOSTEL + 3D Finetuned | **21.358** | **0.8151** | **0.0093** | **29.834** | **18.552** | **0.7251** | **0.0175** | **34.086** | **22.133** | **0.8326** | **0.0083** | **29.174** | **19.790** | **0.7663** | **0.0156** | 40.420 |
| MOSTEL 3D from scratch | **21.256** | **0.7630** | **0.0097** | **28.790** | **18.592** | **0.6266** | **0.0173** | **35.000** | **21.976** | **0.7801** | **0.0084** | **28.639** | 19.698 | 0.6661 | 0.0157 | 42.897 |

Table 3. Quantitative results on Syn3Dtxt-eval-2k, Syn3Dtxt-wrap, Syn3Dtxt-eval-advanced, and Tamper-Syn2k. †means the methods that we reproduced. Best two in each metric column are shown in **Boldface**.

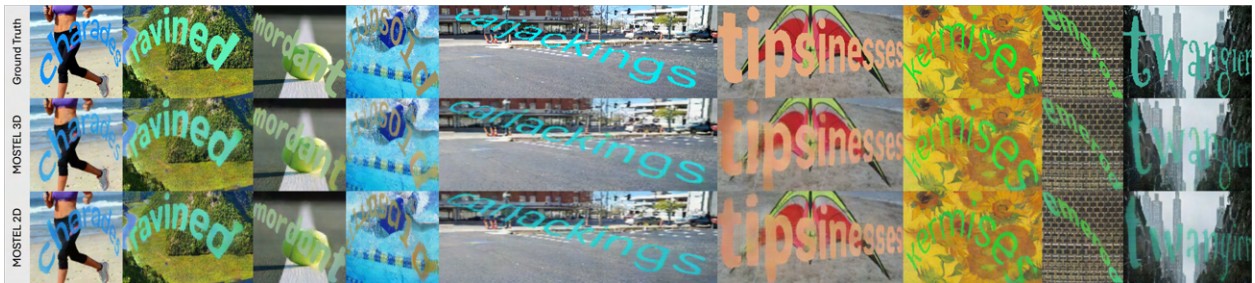

Figure 4. Qualitative Comparison between 2D and 3D models

3. **Curvature Adaptation.**: Finally, the model underwent additional fine-tuning using the Syn3DTxt-wrap dataset to explicitly train on pronounced curvature and complex geometric distortions, enabling robust handling of challenging 3D scenarios.

To facilitate fair comparisons in subsequent experiments, we additionally trained two comparative models. The first comparative model was fine-tuned from the baseline following the above training strategy but employed only binary 2D masks. This approach ensured consistency with traditional 2D methods in terms of data distribution. The second comparative model was trained entirely from scratch using exclusively the Syn3DTxt-150k dataset with 3D masks, serving as an additional benchmark for evaluating our incremental training strategy.

### 4.3. Evaluation Metries

For visual quality assessment, we employ commonly used metrics, including: (i) *SSIM* (Structural Similarity Index Measure), quantifying structural similarity; (ii) *PSNR* (Peak Signal-to-Noise Ratio), measuring image fidelity; (iii) *MSE* (Mean Squared Error), evaluating pixel-level differences; and (iv) *FID* (Fréchet Inception Distance) [5], assessing statistical differences between feature distributions.

### 4.4. Performance Comparison

**Implementation.** We evaluated our trained models across multiple data sets, including Tamper-Syn2k (from MOSTEL [11]), our proposed Syn3DTxt (including the advanced data set), and Syn3DTxt-wrap. Additionally, we compared our model with one GAN-based methods, SRNet [14], and one diffusion-based method, TextCtrl [16], using their provided checkpoints. Quantitative results are presented in Tab. 3, while qualitative comparisons are shown in Fig. 4, Fig. 5 and Fig. 6. Notably, TextCtrl lacks the crucial input required for evaluation on Tamper-Syn2k, limiting its effective comparison on this dataset.

**Text Fidelity in 3D Rotation.** To intuitively demonstrate our method's effectiveness in capturing realistic visual effects during 3D text rotation, we present examples of horizontal rotation (yaw $\phi$) in Fig. 5a and vertical rotation (pitch $\theta$) in Fig. 5b, with differences highlighted by red boxes. When text rotates, regions closer to the viewer visually appear thicker, while those farther away become thinner, creating a clear perspective triangle. To explicitly illustrate this phenomenon, we placed two reference lines on the ground-truth image (second row of Fig. 5a), clearly highlighting the perspective effect induced by rotation. These identical reference lines were also applied to the output images from the two models on the left side for direct visual comparison.

Our results indicate that the model fine-tuned with 3D data accurately captures and preserves the intended 3D perspective features, naturally displaying thicker text in closer regions and thinner text in distant areas, all while maintaining clear glyph structures. In contrast, the model trained exclusively on 2D data fails to adequately

Figure 5. Four visual examples of different models (a) Horizontal 3D Rotation Comparison, Visualization of model outputs under horizontal rotation (rotation along the $\phi$-axis). (b) Vertical 3D Rotation Comparison, Visualization of model outputs under vertical rotation (likely along the $\theta$-axis).

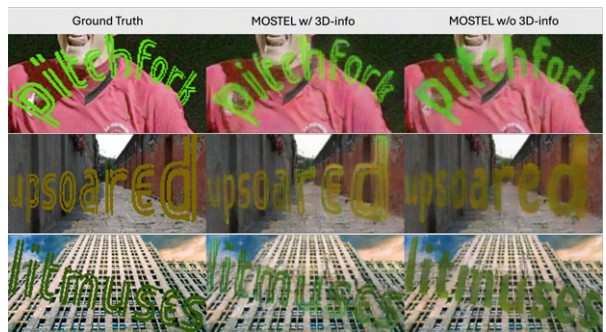

Figure 6. Font Characteristic Preservation Comparison Between 2D and 3D Models

capture these perspective cues, resulting in erosion-like and dilation-like distortions that significantly degrade glyph fidelity, regardless of the text's proximity to the viewer (see Fig. 5a). Moreover, our method effectively retains vertical textual features, as demonstrated in the second row of Fig. 5b, where the 3D-trained model successfully preserves glyph structures under vertical rotation (pitch $\theta$), whereas the 2D-trained model mistakenly transforms the character 'h' into 'n'. Additionally, we observe that the model trained with the 3D dataset demonstrates superior performance in preserving distinctive features of uncommon fonts. (see Fig. 6)

Quantitatively, as shown in Tab. 3, our method achieved an improvement of approximately 10 percentage points across all evaluation metrics, with particularly notable gains in *SSIM* and *FID* (**15%** and **18%**, respectively). The table reports results on four benchmark datasets, Syn3DTxt-eval-2k, Syn3DTxt-wrap, Syn3DTxt-eval-advanced, and Tamper-Syn2k using four widely adopted metrics, PSNR, SSIM, MSE, and FID. The best two results in each metric column are highlighted in **boldface**, clearly demonstrating the consistent superiority of our proposed methods over existing baselines.

## 5. Limitation and Conclusion

**Limitation.** Although our study demonstrates significant improvements by explicitly incorporating 3D geometric characteristic, it still faces challenges when editing text with highly arbitrary shapes and extremely complex curvature. These scenarios involve intricate geometric characteristics that are difficult to fully capture and disentangle, even with detailed 3D representations. Moreover, the original MOSTEL framework does not address surface normal, which introduces additional difficulties in integrating 3D cues effectively into its architecture. While our incremental training strategy enhances model robustness, fully generalizing to arbitrary-shaped text editing remains a key challenge for future research. In addition, the quantitative metrics used in this study, such as *SSIM* and *FID*, are effective in evaluating visual quality and fidelity but primarily assess pixel-level differences or feature similarity in latent space. As such, these metrics may not fully reflect human visual perception of coherence and realism, especially under complex geometric transformations. A more comprehensive, objective evaluation metric aligned more closely with human perception would further benefit the development of scene text editing tasks.

**Conclusion.** This work presents a novel synthetic data generation toolkit and a structured incremental training strategy aimed at progressively integrating complex 3D geometric characteristic into the MOSTEL architecture. By fine-tuning with our proposed Syn3DTxt-150k and Syn3DTxt-wrap datasets, our model achieves significant improvements in capturing realistic perspective features and maintaining glyph structures under challenging 3D rotations. Extensive quantitative experiments and qualitative visual results validate the superiority of our approach, particularly with notable gains in *SSIM* and *FID* metrics. Overall, our findings highlight the importance and effectiveness of detailed 3D geometric encoding for achieving high-quality text editing in realistic and complex visual scenarios.

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
