# OpenReview forum: "$Syn3DTxt:\: Embeding\: 3D\: Cues\: for\: Scene\: Text\: Generation$"
_thecvf.com/CVPR/2025/Workshop/SyntaGen — SyntaGen 2025 Poster_

### Official Review · Reviewer_DE7E · 2025-03-26
**The paper presents a promising approach for enhancing scene-text datasets, though it requires some clarification**

**Rating:** 6
**Confidence:** 3

**Review:**

**Summary**

In the scene text generation task, prior methods for scene text generation mainly use 2D datasets, which lack the geometric details needed to align with real-world 3D environments. In this paper, they propose a synthetic data generation framework to augment 3D geometric cues within the dataset by providing additional RGB-masks. RGB-masks are generated by embedding 3D spatial information into the rendered text images using RGB-encoded surface normals. They evaluate the proposed method for augmenting the dataset using the MOSTEL architecture as a baseline. The training is divided into multiple stages first they follow MOSTEL baseline training setting on 2D and then additional fine-tuning on their augmenting dataset. The evaluation shows that models trained on 3D-augmented datasets outperform those trained on traditional 2D data.

**Strength**
- The paper is well-structured and easy to understand.
- The way of encoding normal vectors within RGB-mask is interesting and show effectiveness. As These normal vectors help the model better understand the interactions between the text and the environment, particularly in terms of perspective projections and spatial relationships.
- Although the two-stage training process produces the best results, the authors show that training from scratch on their 3D data also yields comparable performance.

**Weakness**
- In lines 351-354, the authors mention that they define the rotation distribution to mimic real-world scenarios. However, they do not provide any supporting evidence or explanation for the specific distribution they observed. It would be helpful if they could clarify the rationale behind their rotation distribution and provide more insight into the real-world data they based it on.

---

### Official Review · Reviewer_2GV5 · 2025-03-27
**This paper presents an interesting data generation approach to enhance conventional 2D text by adding 3D geometric cues (surface normals). However, there are some concerns in the Weaknesses section that should be addressed.**

**Rating:** 6
**Confidence:** 3

**Review:**

### **Paper Summary:**
This paper proposes a method to construct 3D geometric text by using surface normals in order to encode the 3D geometric clue as a representation instead of conventional 2D text. The dataset generation process also includes multiple manipulations such as rotation, bending, and scaling to simulate complex real-world text. The experiment shows that fine-tuning with the Syn3DTxt dataset improves performance across all metrics when tested on the Syn3DTxt test set.

### **Paper Strengths:**
1. The paper is well written and easy-to-follow.
2. The idea of using surface normals to encode 3D text orientation and its geometric cues is interesting and has also been shown to be effective in improving results, as demonstrated in Table 1.

### **Paper Weaknesses:**
On the other hand, the paper has the following weeknesses:
1. In Table 1, fine-tuning with 2D and 3D data from the Syn3DTxt dataset helps improve all scores on the Syn3DTxt-eval-2k and Syn3DTxt-wrap test sets. However, on the Tamper-Syn2k dataset, fine-tuning with both 2D and 3D seems to perform worse than using MOSTEL alone. This is unclear about the generalization whether the Syn3DTxt dataset truly help improve performance on existing 2D test sets, or does it only enhance performance on 3D tests derived from the Syn3DTxt dataset?


### **Justification For Recommendation And Suggestions:**
This work introduces an interesting idea to encoded the surface normal into text representing the orientation of it beyond conventional 2D text. However, the paper could be further improved by addresing the concerned in the Weaknesses and additional comments section.

### **Additional Comments For Authors:**
1. In L523, the author mentions that when text is rotated, the region closer to the viewer should visually appear thicker. However, in Figure 5(a), the model without 3D-info seems to produce a visually thicker appearance on the letter "g" even though the text looks more corrupted compared to the version with 3D-info. This example is unclear and could poses some misaligment with the point of having 3D-info data.

---

### Official Review · Reviewer_F6H6 · 2025-03-27

**Rating:** 5
**Confidence:** 3

**Review:**

**Summary**

The paper addresses the problem of Scene Text Editing (STE) by introducing a new synthetic data generation framework and accompanying dataset. The authors identified that most current dataset only have 2D augmentations for text in images. The proposed Syn3DTxt dataset mitigate this by incorporating arc distortion, font transformation, and 3D rotation processing. Experiments on a baseline STE model shows that the dataset can effectively increase performance.

**Strengths**

- The quantitative results show that the proposed dataset can bring significant performance boost to current scene text editing models.

**Weaknesses**

- In line 447 the authors said to have done experiments on the MLT-2017 dataset. I do not see the corresponding section in the paper.
- Only use one model architecture for experiments. While the results of applying Syn3DTxt on MOTEL is impressive, what if we apply it to another model architecture? Will Syn3DTxt provides the same increase?
- All Figures should include the source image with source text. Only showing the source image with the target text (ground truth) makes it hard to see how effective the model is at editing.
- Details about the synthetic data generation are missing. Where were the images obtained? Did the authors crawl the images from the internet or use another existing image dataset?

---

### Decision · Program_Chairs · 2025-03-31

**Decision:**

Accept (Poster)

**Comment:**

The paper presents a method to augment 2D data with 3D cues for effective scene text editing. Two out of three reviewers are positive about the paper, particularly on the strength of using surface normals to encode 3D text. A major concern is that the performance is only improved on the test set of Syn3DTxt. Given the potential benefit of the proposed method, the final decision is to accept the paper. The authors are encouraged to include more baselines, and more experiments to demonstrate the generalizability of the proposed method when testing on other datasets.